# A vocabulary of ancient peptides at the origin of folded proteins

**Vikram Alva, Johannes Söding[†], Andrei N Lupas\***

Department of Protein Evolution, Max Planck Institute for Developmental Biology, Tübingen, Germany

**Abstract** The seemingly limitless diversity of proteins in nature arose from only a few thousand domain prototypes, but the origin of these themselves has remained unclear. We are pursuing the hypothesis that they arose by fusion and accretion from an ancestral set of peptides active as co-factors in RNA-dependent replication and catalysis. Should this be true, contemporary domains may still contain vestiges of such peptides, which could be reconstructed by a comparative approach in the same way in which ancient vocabularies have been reconstructed by the comparative study of modern languages. To test this, we compared domains representative of known folds and identified 40 fragments whose similarity is indicative of common descent, yet which occur in domains currently not thought to be homologous. These fragments are widespread in the most ancient folds and enriched for iron-sulfur- and nucleic acid-binding. We propose that they represent the observable remnants of a primordial RNA-peptide world.

**\*For correspondence:** andrei.lupas@tuebingen.mpg.de

**Present address:** [†]Research Group for Quantitative and Computational Biology, Max Planck Institute for Biophysical Chemistry, Göttingen, Germany

**Competing interests:** The authors declare that no competing interests exist.

## Introduction

The origin of most present-day proteins can be attributed to combinatorial shuffling and differentiation events involving a basic set of domain prototypes, which act as the unit of protein evolution (*Anantharaman et al., 2001*; *Apic et al., 2001*; *Ponting and Russell, 2002*; *Orengo and Thornton, 2005*). Many of these domains can be traced back to the time of the Last Universal Common Ancestor (LUCA) (*Kyrpides et al., 1999*; *Koonin, 2003*; *Ranea et al., 2006*), a hypothetical primordial organism from which all life on earth is thought to have descended approximately 3.5 billion years ago (*Glansdorff et al., 2008*). The origin of domains themselves, however, is poorly understood. An origin de novo, by random concatenation of amino acids, appears impossible due to the high sequence complexity and low folding yield of polypeptides, as well as to the absence of abiotic processes that could produce peptide chains of more than 5–10 residues. The abiotic scenario would also leave open the fundamental question as to how the information contained in successful polypeptides could have been passed on.

Many lines of evidence, including the identification of local sequence and structure similarity within domains of different fold (*Brennan and Matthews, 1989*; *Doherty et al., 1996*; *Copley et al., 2001*; *Grishin, 2001b*; *Friedberg and Godzik, 2005*; *Alva et al., 2007*; *Andreeva et al., 2007*), or the frequent construction of domains by repetition of subdomain-sized fragments (*McLachlan, 1987*; *Andrade et al., 2001*; *Hocker et al., 2002*; *Chaudhuri et al., 2008*; *Remmert et al., 2010*), show that domains might not constitute the only evolutionary unit of protein structure. These observations led to the proposal that the first folded domains arose by repetition, fusion, recombination, and accretion from an ancestral set of peptides (*Fetrow and Godzik, 1998*; *Lupas et al., 2001*; *Soding and Lupas, 2003*) that emerged in the RNA world (*Gilbert, 1986*), in which RNA served both as carrier of genetic information and catalyst of metabolic reactions (*Jeffares et al., 1998*). According to this model, the local similarities found in modern proteins represent the observable remnants of such peptides. In the RNA world, which is widely thought to have

**eLife digest** Life as we know it today is largely the result of the chemical activity of proteins. Much research suggests that the ancestors for most modern proteins were already present in the 'Last Universal Common Ancestor', a theoretical ancient organism from which all life on earth descended and which lived around 3.5 billion years ago.

Today, related versions of these ancestral proteins are found in organisms as different as bacteria, humans and plants. While they seem highly diverse, these proteins were all assembled from only a few thousand modular units, termed domains. However, it is not clear how the first domains emerged.

Previously, in 2001 and 2003, researchers hypothesized that the first protein domains arose by joining and swapping short lengths of proteins called peptides that had emerged before there were living cells on earth – a time that is often called the "RNA world". Now, Alva et al. – including the researchers involved in the 2003 work – have attempted to detect remnants of these ancient peptides in modern proteins.

Alva et al. first compared modern proteins in a way that is similar to how linguists have compared modern languages to reconstruct ancient vocabularies. This revealed 40 fragments that occur in seemingly unrelated proteins, but are very similar in their sequence and structure. These fragments are commonly found in what are likely the oldest observable proteins, and are involved in the activities that are most fundamental to life (for example, binding to DNA and RNA). This led Alva et al. to propose that these fragments represent the observable remnants of a primordial "RNA-peptide world".

The hypothesis that proteins evolved from peptides provides a number of predictions that can be tested in experiments. These fragments open avenues to explore in the laboratory the origin of modern proteins and to build new proteins not seen in nature.

been an important intermediate stage in the origin of cellular life, simple peptides may have been recruited by RNA to expand its functional repertoire. The catalytic range of RNA molecules is restricted (*Joyce, 2002*) and peptides are good chelators of metals and small molecules. Peptides are also beneficial for RNA thermostability and folding specificity, and for the formation of oligomeric complexes. For these reasons, peptides of initially abiotic origin may have been co-opted as cofactors. In time, selective pressures on availability, interaction specificity, and functional effectiveness would have driven the emergence of longer, RNA-encoded and -produced peptides, optimized for the formation of secondary structure by exclusion of water with the RNA scaffold. It is known that many peptides formed of the 20 proteinogenic amino acids have an intrinsic affinity for RNA and form secondary structures upon binding (*Patel, 1999*; *Das and Frankel, 2003*). By repetition, fusion, recombination, and accretion, these preoptimized peptides would have reached a level of complexity enabling them to exclude water by hydrophobic contacts, making them independent of the RNA scaffold. In this theory, protein folding would have been an emergent property of peptide-RNA coevolution.

## Results and discussion

### A comparative approach to domain evolution

Given the striking conservation of many proteins in sequence and structure across evolutionary time, we conjectured that if this hypothesis is true, we might be able to see remnants of these primordial peptides in modern proteins. To this end, we decided to take a comparative approach based on the systematic analysis of present-day domains, similar to the approach taken by linguists to reconstruct ancient vocabularies by comparing modern languages. The comparative studies of languages and of proteins, in fact, exhibit many parallels because of their shared linear nature of information storage, the high conservation of evolutionary modules, and the similarity of evolutionary constraints acting upon them (*Gray and Atkinson, 2003*; *Pagel et al., 2007*; *Searls, 2013*; *List et al., 2014*) (*Figure 1A*). Today, for instance, it is known that hundreds of words in European languages contain

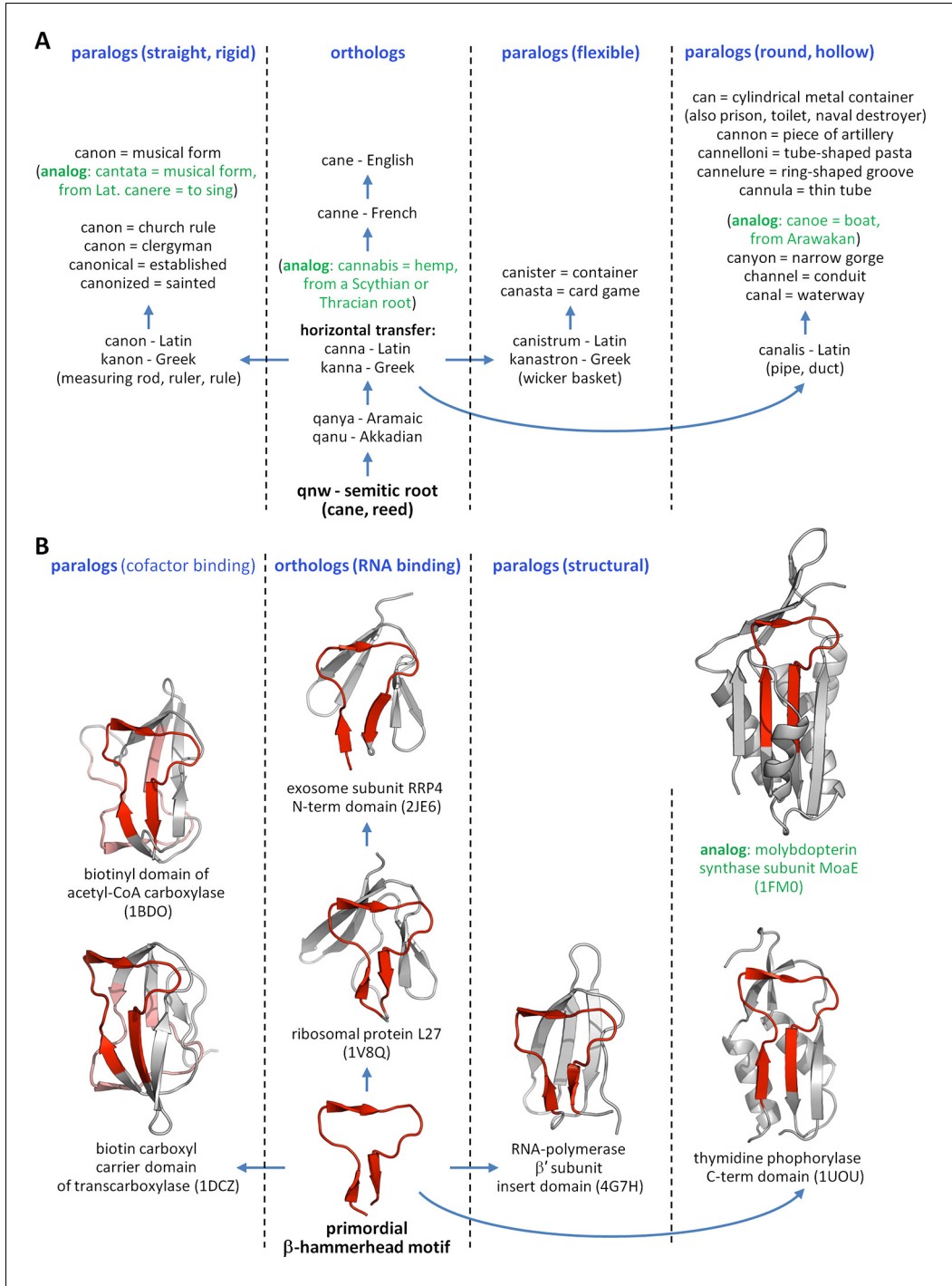

**Figure 1.** The evolution of words and proteins shows many parallels. (**A**) The Semitic root *qnw (*qanaw-)*, meaning reed, is the ancestor of hundreds of words in many different languages, following the same mechanisms as already known from biological evolution. Here we track the descendants of this root in the English language, arisen through the intermediary of Latin and Greek. In addition to the orthologous Greek word *kanna* (reed), paralogous cognates arose in antiquity based on certain attributes of reed, e.g., the levelling rule *kanon* (taking the straight and rigid attribute of reed), the wicker basket *kanastron* (flexible), and the Latin water duct *canalis* (round and hollow). A few examples of analogous words, which appear to be related to the descendants of *qnw* but have different evolutionary origins, are shown in green. (**B**) The primordial β-hammerhead motif (shown in red) is seen in four different folds, which cover a wide array of functions. Following our hypothesis of an origin in the RNA world, we propose that RNA binding is the orthologous function of this peptide, seen today in ribosomal protein L27 and

*Figure 1 continued on next page*

*Figure 1 continued*

exosome subunit RRP4. Paralogous functions arose around the time of the Last Universal Common Ancestor from its ability to form a biotin-binding domain by duplication, yielding the biotin-dependant enzymes of the barrel-sandwich hybrid fold, and to serve as a structural element in domains formed by accretion, yielding a domain of RNA-polymerase β' subunit, as well as a range of enzymes with an α/β-hammerhead fold. By our analysis, enzymes classified in the α/β-hammerhead fold superfamily d.41.5, such as MoaE, are analogous to the other superfamilies in this fold, due to a lack of detectable sequence similarity, but nevertheless contain a supersecondary structure resembling the β-hammerhead.

the conserved Semitic root qnw (*qanaw-) meaning 'reed' (*Huehnergard, 2011*), despite it having diversified into a wide range of functional forms by the same processes as already familiar from biological evolution (e.g., orthology, paralogy, horizontal transfer).

To reconstruct the 'vocabulary' of ancient peptides, we aimed at finding local similarities in sequence and structure within globally different folds, which are presently thought to have arisen independently, by convergent evolution. Since the events that led to the emergence of domains took place before the time of LUCA, modern domains might only retain weak signals thereof in their sequences. As protein structures diverge much more slowly than their sequences, structural similarity is often used for identifying such distant events. However, similar structures may have arisen convergently, owing to the limited number of conformations available to a folded polypeptide chain, particularly at the supersecondary structure level (*Salem et al., 1999*; *Kim et al., 2009*; *Fernandez-Fuentes et al., 2010*). Consequently, structure similarity alone does not provide conclusive evidence of common ancestry. In contrast, the combinatorial sequence space is enormous and many sequences are compatible with a particular local structure, so that sequence convergence is rare. Thus sequence similarity is a more reliable marker for common ancestry (*Soding, 2005*; *Kim et al., 2009*). We have therefore employed sequence similarity, as evaluated by the comparison of profile hidden Markov models (HMMs) (*Soding et al., 2005*), as a first criterion for inferring common ancestry of domains in this study. Due to the large evolutionary divergence of sequences, we used structural similarity as a second criterion to confirm the potential homology relationships.

## Reconstructing a vocabulary of primordial peptides

To implement this comparative approach between domains of different fold, we needed a reference database for the assignment of domains to fold types. For this we used the SCOPe (*Murzin et al., 1995*; *Fox et al., 2014*) database (release 2.03), which is a point of reference in the classification of protein folds. In this database, the first two classification levels (family and superfamily) capture homologous relationships, while the grouping of structurally similar superfamilies into the same fold reflects convergent evolution, i.e. analogy. In order to reduce the very large background of obvious homologous matches, we filtered SCOPe to a maximum of 30% sequence identity (SCOPe30). At this level, many relationships considered homologous by SCOPe are removed, whereas representatives for all families and superfamilies are still retained. As detailed in the 'Materials and methods', we then compared the resulting domain set in sequence space using HHsearch with stringent settings, and subsequently in structure space using TM-align (*Zhang and Skolnick, 2005*). We plotted the obtained scores separately for comparisons within families (presumed homologous relationships; *Figure 2A*) and between folds (presumed analogous relationships; *Figure 2B*). The expected score distributions would have been at the top, right (high HHsearch probabilities, high TM-scores) for homologous relationships and bottom, left (low HHsearch probabilities, low TM-scores) for analogous ones. In fact, the distributions we obtained were bimodal, with a higher incidence of scores at both the top, right and bottom, left of the plots. The presence of some low-scoring relationships in the homologous set (*Figure 2A*) did not appear too surprising, given the considerable evolutionary distance of some relationships captured in SCOPe, but the presence of some high-scoring relationships in the analogous set (*Figure 2B*) did, as it suggested the presence of hitherto unrecognized homologous relationships. We decided to explore these further, using cut-offs at HHsearch probabilities of 70% (corresponding to P-values < 5e-05) and TM-scores of 0.5. At these threshold values, about a fourth of presumed homologous relationships in SCOPe30 and >99.95% of matches between folds, which are presumed to be analogous, are omitted. This provided a substantial

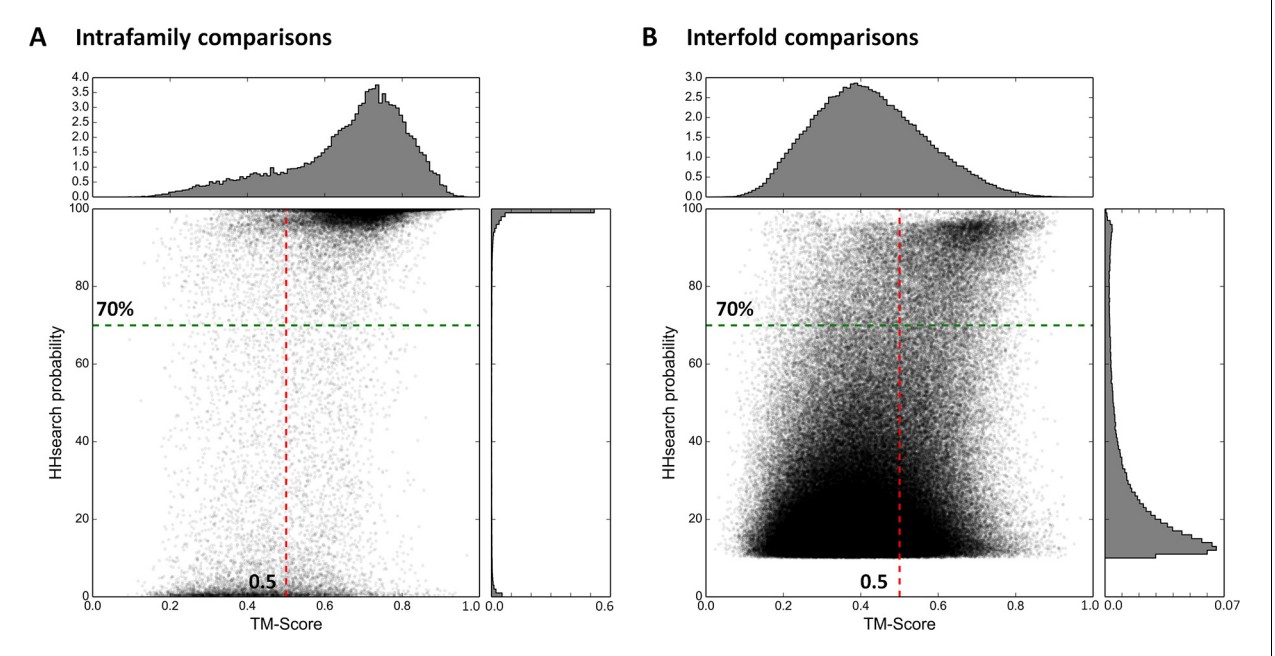

**Figure 2.** Estimation of cut-offs for HHsearch probability and TM-score. We compared all domains in the SCOPe30 set in sequence space using HHsearch and subsequently in structure space using TM-align (see 'Materials and methods'), and plotted the obtained scores. Separate plots for comparisons of domains within families (**A**) and between folds (**B**) were generated. Scores would have been expected at high HHsearch probabilities and TM-scores for intrafamily comparisons (presumed homologs, Panel A) and low HHsearch probabilities and TM-scores for interfold comparisons (presumed analogs, Panel B), but the score distributions were in fact bimodal, as also illustrated by the histograms top and right in each panel, which are plotted as probability density functions. In the comparison of domains of different fold, matches with an HHsearch probability of < 10% are not plotted.

margin of safety in evaluating the high-scoring relationships between domains presumed to be analogous (*Figure 2B*).

Since the comparisons made in the analogous set were between domains of different fold, the high-scoring matches always involved subdomain-sized fragments. Those fragments that satisfied the cut-offs and a number of additional criteria, as detailed in the 'Materials and methods', were clustered together automatically if they overlapped by at least 80% of their length. Following upon this step, clusters were inspected individually and merged further where appropriate, as described in the 'Materials and methods'. We obtained 65 clusters, of which 20 relied on a global similarity of the folds and were omitted from further consideration. Such cases of globally similar folds that are nevertheless classified as different are mainly due to homologous fold change events, such as strand invasion, hairpin swapping, or circular permutation (*Grishin, 2001a*; *Andreeva and Murzin, 2006*; *Alva et al., 2008*). Thus, for instance, SCOPe superfamilies d.12.1 and d.58.7 are clearly related globally by circular permutation, despite being classified into different folds. A further 5 groups were due to artifacts in domain boundaries assigned by SCOPe and were also omitted. For example, the d.79.4.1 family makes connections to the a.5.10.1 family, as its sequence in SCOPe encompasses the latter.

This yielded a final set of 40 clusters, within which all fragments were inspected individually, superimposed, and trimmed to a consensus length where appropriate (*Figure 3*, *Figure 3—source data 1*; *Table 1*). As examples, we show the different embodiments of the β-hammerhead motif, found in 4 folds comprising 8 superfamilies (*Figure 1B*, *Figure 3*: fragment 12), and the nucleic acid-binding helix-hairpin-helix motif (*Doherty et al., 1996*; *Shao and Grishin, 2000*), found in 8 folds comprising 15 superfamilies (*Figure 3*: fragment 2, *Figure 4*). The median length of the 40 consensus fragments is 24 residues, with the shortest fragment comprising 9 and the longest 38 residues. This is in accord with the expectation that ancient peptides were simple and subdomain-sized. Of these fragments, only about half correspond to typical supersecondary structure elements, such as

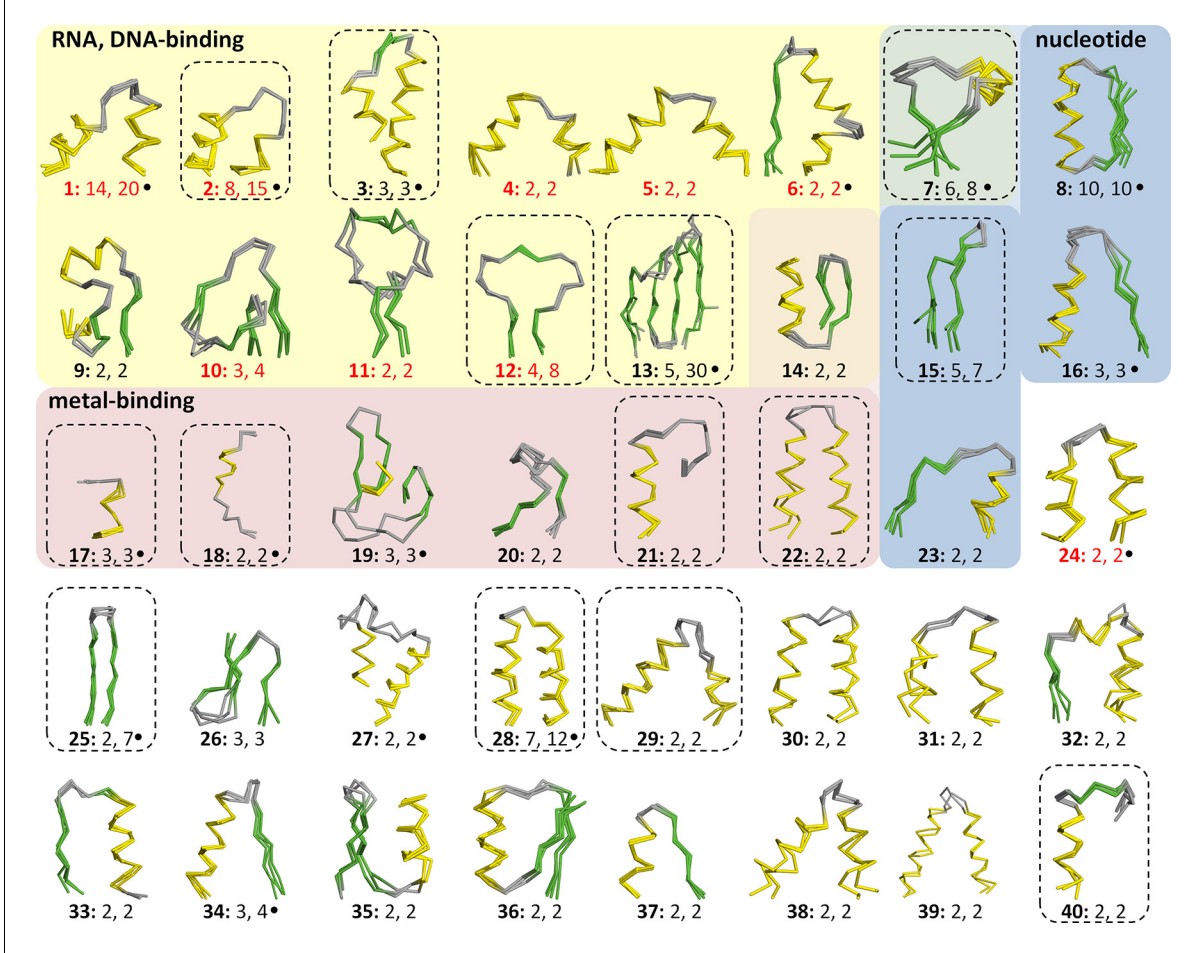

**Figure 3.** Vocabulary of primordial peptides that gave rise to folded proteins. The 40 peptides we detected are shown as ensembles in backbone representation; α-helices are coloured in yellow, β-strands in green, and loops in gray. Detailed information on each fragment is provided in *Table 1* and *Figure 3—source data 1*. The fragments are numbered sequentially and their occurrence in different folds and superfamilies of SCOPe is given. Fragments reported individually before are indicated by a dot. Nucleic-acid binding, nucleotide-binding, and metal-binding motifs are highlighted in yellow, blue, and red, respectively. Fragments found in ribosomal proteins are indicated by red font colour. Fragments that form folds by repetition are boxed.

The following source data and figure supplement are available for figure 3:

**Source data 1.** Multiple sequence alignments and accession details for the 40 primordial fragments shown in *Figure 3* and the 5 B-set fragments shown in *Figure 3—figure supplement 1*.

**Figure supplement 1.** Vocabulary of primordial peptides (B-set).

α-hairpins, β-hairpins, β-meanders, and βαβ-elements (*Salem et al., 1999*), whereas the others are unusual fragments with odd shapes, which frequently do not form compact structures. This supports the notion that they predate the emergence of hydrophobic contacts as a driving force for protein folding and that their open structures reflect the association with an RNA scaffold, as still seen in ribosomal proteins today (*Soding and Lupas, 2003*).

To evaluate whether the sequence similarity exhibited by these fragments could be the result of biophysical constraints, rather than common descent, we searched SCOPe30 with each fragment for structurally similar matches: 36 of the 40 had at least one match to another superfamily with a TM-score $\geq$ 0.5, but undetectable sequence similarity (HHsearch probability <10%). Indeed, of these 36 fragments, 34 had more than half of their structure matches to fragments with undetectable sequence similarity. This shows that essentially every one of the structures we detected can be

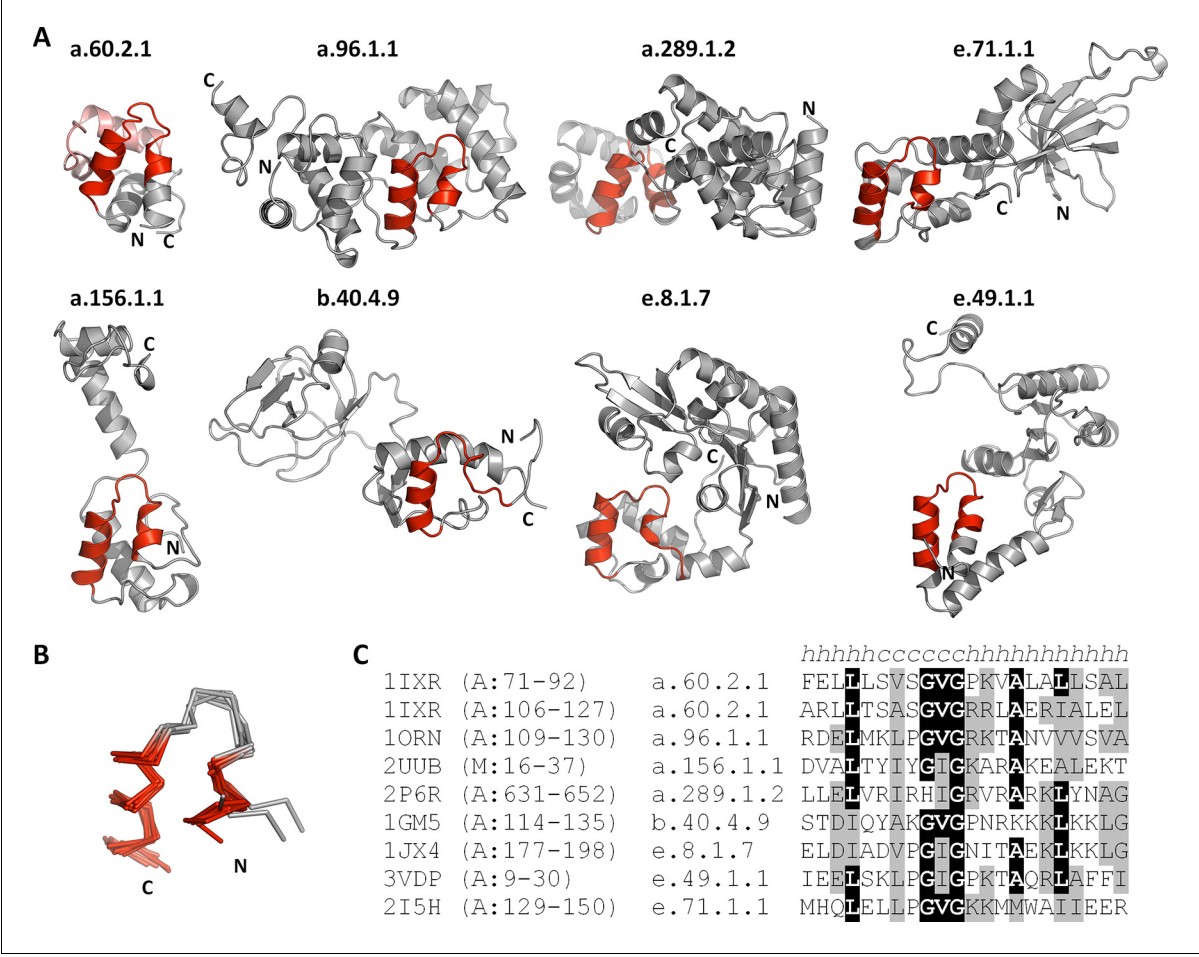

**Figure 4.** The nucleic-acid binding helix-hairpin-helix motif is found in 8 different folds comprising 15 superfamilies. (**A**) Representative domains from the eight SCOPe folds. The motif is coloured in red and the remainder of the structure is shown in gray. The SCOPe family a.60.2.1 contains two copies of this motif, whereas the remaining folds contain one copy each. (**B**) Structural superimposition of the helix-hairpin-helix motifs displayed in panel A. (**C**) Sequence alignment of the motifs shown in panel A. Residues conserved in at least half of the aligned sequences are highlighted in black and similar residues are highlighted in gray.

formed by fundamentally different sequences. To verify this anecdotal observation in a systematic way, we analyzed the relationship between sequence and structure similarity in our fragments by comparison with a reference set of 40 of the most frequent supersecondary structure elements from the Smotifs library (*Fernandez-Fuentes et al., 2010*) (assembled as described in the 'Materials and methods'). We reasoned that a correlation should be detectable in homologs, as these would have started ancestrally with identical sequences and structures, before gradually diverging towards a baseline of similarity. There should be no correlation however in analogs, unless structural constraints had limited the number of residues allowed at specific positions, causing a convergence of the sequences (*Remmert et al., 2010*; *Kopec and Lupas, 2013*). We therefore computed sequence similarity scores using structure-based sequence alignments for each fragment in the two sets against all domains in SCOPe30. For fragments in our set, we considered matches to other superfamilies in which we had detected the respective fragment (listed in the *Figure 3—source data 1*) as homologous and to all other folds as analogous (*Figure 5A*). For the Smotifs fragments, we considered matches to the same superfamily as homologous and to all other folds as analogous (*Figure 5B*). For both fragment sets, the presumed homologous matches show a strong correlation between sequence and structure similarity. The correlation for the Smotifs reference set is expected, as the homologs there follow the generally accepted criteria laid out in the SCOPe classification. Observing a nearly equivalent correlation for our fragment set therefore underscores our inference of

**Table 1.** Data on the 40 primordial fragments.

| Fragment | Number of folds, superfamilies | Repetition in SCOPe foldor superfamily | Ribosomal (SCOPe id; protein) | Ligands | 9 most ancient and basal folds (SCOPe id) | Occurrence in the 10 folds with the largest number of superfamilies in SCOPe |
|---|---|---|---|---|---|---|
| 1[+] | 14, 20 | | a.4.5; S19e | DNA | a.4 | |
| 2[+] | 8, 15 | a.60.2 | a.156.1; S13 | DNA, RNA | | a.60 |
| 3[+] | 3, 3 | a.174.1 | | DNA | c.37.1.20 | |
| 4 | 2, 2 | | a.2.2; L29 | RNA | | a.2 |
| 5 | 2, 2 | | d.14.1; S5 | RNA | | |
| 6[+] | 2, 2 | | d.52.3; S3 | DNA, RNA | | |
| 7[+] | 6, 8 | all folds containing it | | DNA, CTP, SAM, FMN | | |
| 8[+] | 10, 10 | | | FAD, NAD, NEA, NAJ, NAP, NAI, NDP, AMP, NMN, APR, LNC, A3D, ODP, UMA, SAH, SAM, COA, | c.2, c.66 | |
| 9 | 2, 2 | | | DNA | | |
| 10 | 3, 4 | | d.66.1; S4 | RNA | | |
| 11 | 2, 2 | | b.53.1; TL5 | RNA | | |
| 12 | 4, 8 | b.84.1 | b.84.4; L27 | RNA | | |
| 13[+] | 5, 30 | all folds containing it | | DNA | | |
| 14 | 2, 2 | | | ZN, DNA | | |
| 15 | 5, 7 | d.37.1 | | FMN, AMP, FAD | | |
| 16[+] | 3, 3 | | | ATP, GTP, ADP, CTD, DGP, DCP, ANP, TMP, GDP | c.37 | |
| 17[+] | 3, 3 | a.138.1 | | HEM, HEC | | a.24 |
| 18[+] | 2, 2 | a.1.2, d.58.1 | | SF4($Fe_4S_4$) | d.58.1 | d.58 |
| 19[+] | 3, 3 | | | ZN | | |
| 20 | 2, 2 | | | FES | | |
| 21 | 2, 2 | a.39.1, a.139.1 | | CA | | |
| 22 | 2, 2 | a.25.1 | | FEC, FE, FE2, HEM | | |
| 23 | 2, 2 | | | COA | | |
| 24[+] | 2, 2 | | d.45.1; L7/12 | | | |
| 25[+] | 2, 7 | f.4 | | | | |
| 26 | 3, 3 | | | | | g.3 |
| 27[+] | 2, 2 | | | | | |
| 28[+] | 7, 12 | a.118.8 | | | | a.24, a.118 |
| 29 | 2, 2 | b.34.4 | | | | b.34 |
| 30 | 2, 2 | | | | | a.2 |
| 31 | 2, 2 | | | | | |
| 32 | 2, 2 | | | | c.55.1 | |
| 33 | 2, 2 | | | | | |
| 34[+] | 3, 4 | | | | | |
| 35 | 2, 2 | | | | d.58.49 | d.58 |
| 36 | 2, 2 | | | | c.55.5 | |
| 37 | 2, 2 | | | | | |
| 38 | 2, 2 | | | | | a.60 |
| 39 | 2, 2 | | | | | a.118 |
| 40 | 2, 2 | a.20.1 | | | | |

[+]previously reported fragments

homology. Note that, although the correlation is slightly weaker in our fragments than in the Smotifs, this is likely due to the fact that comparisons in our set are made between superfamilies, whereas in the Smotifs they are made within superfamilies, i.e. across smaller evolutionary distances. In contrast, the correlation between sequence and structure similarity in the presumed analogous matches is very weak for both our fragments and the Smotifs. This shows that structurally induced sequence convergence is very low and the fragments can be formed by a broad range of different sequences. Biophysical constraints can therefore not explain the sequence similarity exhibited by our fragments.

We are of course aware that molecular homology cannot be proven rigorously by scientific standards and that the boundaries of what constitutes evidence of common descent evolve continuously as extrapolation from increasingly distant connections are found to yield useful structural and functional predictions. Due to the conservative nature of our sequence comparisons and the range of additional criteria we applied in order to eliminate potentially spurious matches, we expect that we did not extend these boundaries substantially beyond what is already established in the field. Given the depth of analysis in many structural studies, we therefore surmised that at least some of our fragments should have been noted previously. In fact, 40% (16) have been described individually before and connected to deep evolutionary events (*Figure 3*, indicated by a dot; *Table 1*), showing that we are moving within the boundaries of established sequence-structure analyses. To our knowledge, our fragments comprise all but one previously reported case. We failed to detect the ASP-box (*Figure 3—figure supplement 1*: B1), found in six different folds (*Copley et al., 2001*), owing to the stringency of our search criteria (if the HHsearch probability cut-off is lowered to 60%, this fragment is included, along with four others). Examples of previously described fragments include the dinucleotide-binding β-α-β motif (*Buehner et al., 1973*; *Rossmann et al., 1974*; *Wierenga et al., 1986*; *Dym and Eisenberg, 2001*) (*Figure 3*: 8), the KH motif of type I and type II KH domains (*Grishin, 2001b*) (*Figure 3*: 6), the helix-hairpin-helix motif of nucleic acid-binding domains (*Doherty et al., 1996*; *Shao and Grishin, 2000*) (*Figure 3*: 2, *Figure 4*), the EF-Tu-binding α-hairpin of elongation factor EF-Ts and ribosomal protein L7/12 (*Wieden et al., 2001*) (*Figure 3*: 24), and the P-loop of PEP carboxykinases and P-loop NTPases (*Walker et al., 1982*; *Matte et al., 1996*) (*Figure 3*: 16). For many of these fragments, our systematic approach, the growth of structure databases, and improved sequence comparison methods allowed us to identify new instances of occurrence. For example, another occurrence of the P-loop is in the catalytic domain of MurD-like peptide ligases (SCOP c.72.2), where it is involved in binding the phosphate group of a mononucleotide as well.

## Primordial peptides in an RNA world

If our starting assumption of ancestral peptides in the context of an RNA world is correct, some of the most basic functions would be nucleic-acid binding and catalysis. We would therefore expect these properties to be enriched in our set. Indeed, we find that about a third of our fragments make contact with nucleic acids (13), based on evidence from known structures (fragments with a yellow background in *Figure 3*; *Table 1*). For comparison, the Smotifs reference set yielded only 2 nucleic-acid binders, even though it covers about three times as many superfamilies and folds as our fragment set (see 'Materials and methods'). Our nucleic acid-binding fragments include four of the six most highly represented fragments in our dataset, particularly the helix-turn-helix motif (*Sauer et al., 1982*; *Pabo and Sauer, 1984*; *Brennan and Matthews, 1989*; *Suzuki and Brenner, 1995*; *Aravind et al., 2005*) (*Figure 3*: 1- found in 14 folds and 20 superfamilies, abbreviated in the following as 14, 20) and the helix-hairpin-helix motif (*Figure 3*: 2- 8, 15; *Figure 4*). A special case of nucleic-acid interaction is provided by ribosomal proteins, which contain 8 of our 13 nucleic acid-binding fragments (indicated by red font color in *Figure 3*; *Table 1*). Ribosomal proteins are likely to represent the oldest proteins observable today and, for the most part, still require the RNA scaffold to become structured, providing a window into the time when protein domains were being established (*Hsiao et al., 2009*). For comparison, of the nine folds proposed to be the most ancient (*Caetano-Anolles et al., 2007*), based on the comparative analysis of proteomes from diverse branches of life, six encompass at least one of our 40 fragments (*Table 1*).

The second basic function peptides would plausibly have had in the RNA world would have been catalytic, as coordinators of metals, iron-sulfur clusters, nucleotides, and nucleotide-derived cofactors (coenzyme A, NAD(P), FAD), all of which are thought to already have played an essential role in prebiotic metabolism (*White, 1976*; *Wachtershauser, 1992*). Here, however, an enrichment relative

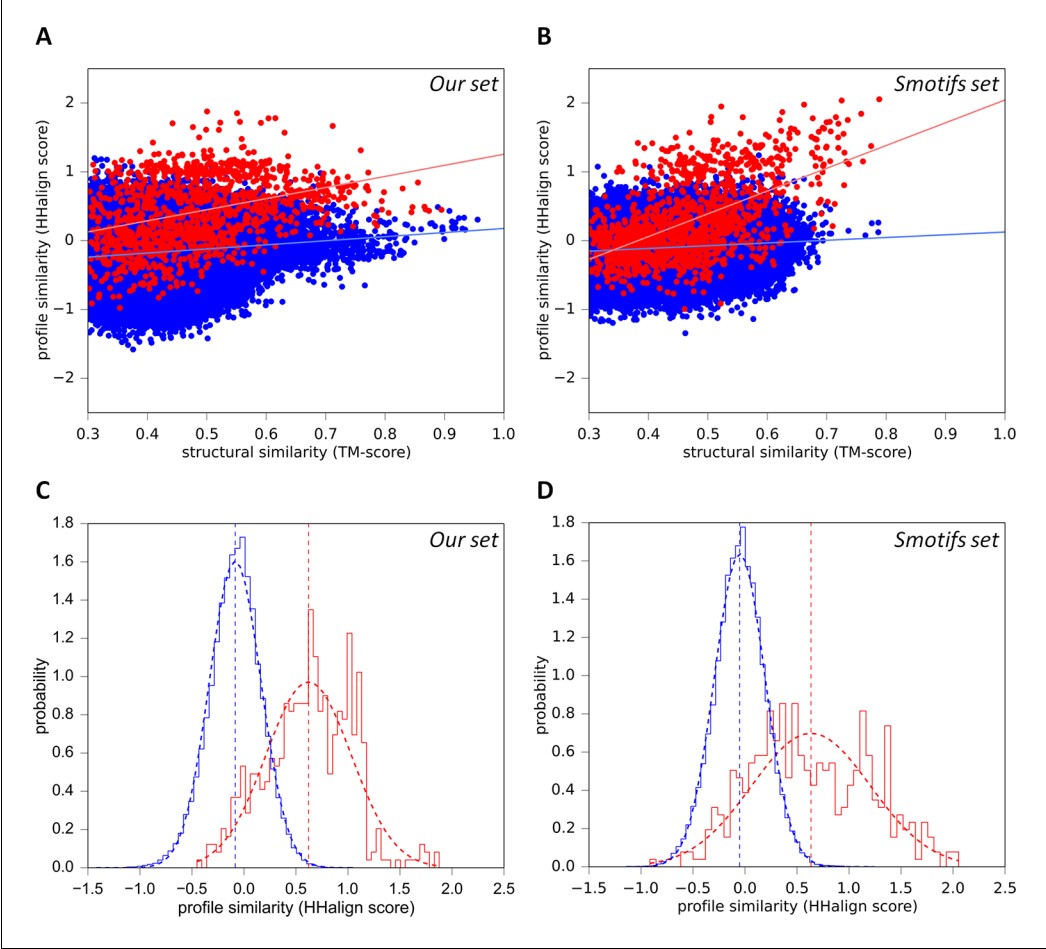

**Figure 5.** Sequence similarity of our fragments cannot be explained by structural constraints. (**A**) For each occurrence of any of our 40 fragments, we searched for structural matches in SCOPe30 and plotted the TM-align score versus the profile-similarity score for the fixed alignment given by TM-align. The putatively homologous matches to occurrences of the same fragment in another superfamily are shown in red. Matches to fragments outside the list of folds in which the query fragment was found to occur (i.e. non-homologous matches) are blue. (**B**) Same as A, but using the Smotifs reference fragments as queries instead of our set. Matches within superfamilies (homologs) are shown in red, matches between fragments from different folds (analogs) are shown in blue. For both sets, sequence and structure similarity scores are significantly correlated for presumably homologous matches (our set: r=0.38; Smotifs: r=0.56, see linear regression lines) but not for analogous matches (our set: r=0.14; Smotifs: r=0.12). (**C, D**) Distribution of profile similarity scores for matches with a TM-score $\geq$ 0.5, for the homologous and analogous distribution in the plots (A) and (B), respectively. The means of the Gaussian fits are exactly the same in C and D.

to the Smotifs reference set is less clearly apparent, being primarily seen for iron-sulfur clusters which are absent in the reference set. Seven of our fragments coordinate metal ions and iron-sulfur clusters (red background in *Figure 3*; *Table 1*), e.g., the cytochrome-heme-attachment motif (*Figure 3*: 17) (*Mathews et al., 1985*) and the 4Fe-4S coordinating peptide (*Figure 3*: 18) (*Lupas et al., 2001*), and five bind nucleotides and nucleotide-derived cofactors (blue background in *Figure 3*; *Table 1*). Particularly two of these fragments, the P-loop motif (*Figure 3*: 16) and the dinucleotide-binding β-α-β motif (*Figure 3*: 8), are at the core of some of the largest enzyme superfamilies in nature and play a central role throughout metabolic processes. We note that in all these fragments, the contribution of the peptides is the coordination of catalytic cofactors and not the provision of catalytic residues *per se*, in accordance with a primordial role as cofactors of RNA-driven catalysis. Thus, for example, the role of the P-loop in nucleotidases is to coordinate the nucleotide, the mechanism for its hydrolysis having evolved independently in different lineages of this superfamily.

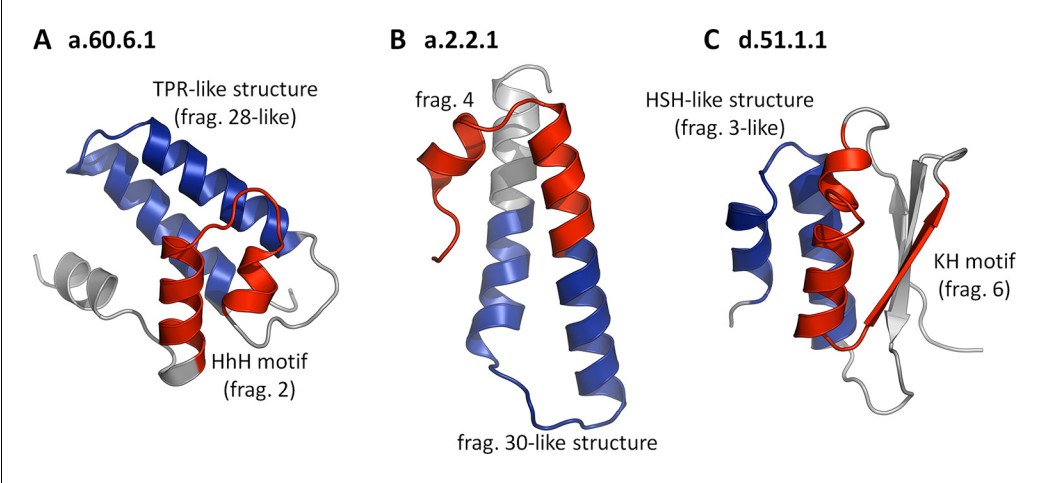

**Figure 6.** Folds showing two nonidentical fragments, one of which is which is not significant by our criteria. No SCOP fold combines two of our fragments at the cutoffs used in this study (TM-score ≥ 0.5 and HHsearch probability ≥ 70%). If we however omit the sequence cutoff entirely for the second fragment, combinations become apparent. In these three examples, the 'significant' fragments are colored in red, the 'nonsignificant' ones in blue, and the remainder in gray. The structures are: (**A**) a.60.6.1, N-terminal domain of polymerase β (4KLI, A: 10-91), (**B**) a.2.2.1, 50S ribosomal protein L29 (1VQ8, V: 1-65), and (**C**) d.51.1.1, KH domain-like hypothetical protein APE0754 (1TUA, chain A: 1-84).

## Repetition as a dominant force in the origin of folds

When we initiated this study, we believed that assembly from non-identical fragments may have been one of the primary forces in the evolution of domains (*Lupas et al., 2001*; *Soding and Lupas, 2003*), and we expected to find many examples demonstrating it. However, we did not find even one domain that contained two or more different fragments from our set. Our fragments either form their folds by repetition or in single copy, decorated by heterologous structural elements. At present, we find the reasons for the lack of fragment combinations unclear, but we note that many of our fragments might represent dominant cores that guide the folding of the remainder of the polypeptide chain (*Religa et al., 2007*) and would, as such, not be generally compatible with each other. In the few cases where they would be sufficiently compatible to produce an initial fold capable of entering biological selection, they would be under considerable pressure to adapt to the new structural environment. This, in most cases, might lead to the retention of only one dominant fragment, the other(s) adjusting by rapid divergence, making them undetectable by our methods. For an initial exploration of this possibility, we asked whether any fold in SCOPe30 could be found to contain two or more of our fragments if we relaxed the sequence similarity requirement for all but one of them. We analyzed in detail the largest family from each superfamily containing one of our fragments (188 in all) for such combinations of 'significant' and 'non-significant' fragments. As long as we retained any sequence similarity cutoff for the 'non-significant' fragments, even as low as HHsearch probabilities of 10%, no combinations could be found. If we however removed the sequence similarity requirement entirely for the 'non-significant' fragments, asking only for structural similarity (TM-score ≥ 0.5), over 50% of the families showed fragment combinations. In *Figure 6*, we have collected examples, in which the fold is substantially formed by the combination of a 'significant' with a 'nonsignificant' fragment, showing that the incompatibility we observe for our fragments is not of a geometrical nature.

While we were unable to detect fragment combinations, repetition is wide-spread, seen for 14 of our 40 fragments (35%) (indicated by a dotted box in *Figure 3*; *Table 1*). Nine of these are also capable of forming folds in single copy, with additional decoration. For example, the TPR element (*Figure 3*: 28) is found in multiple copies in domains belonging to the TPR-like superfamily (a.118.8) (*D'Andrea and Regan, 2003*), but only in single copy in other folds that contain it (*Figure 7A*). Also, the β-hammerhead motif (*Figure 1B*, *Figure 3*: 12), which resembles a hammerhead, is duplicated in the barrel-sandwich hybrid fold (b.84), but occurs in single copy in the α/β-hammerhead fold (d.41) and in two other folds (e.29 and f.46). Two further fragments also occur in variable numbers per

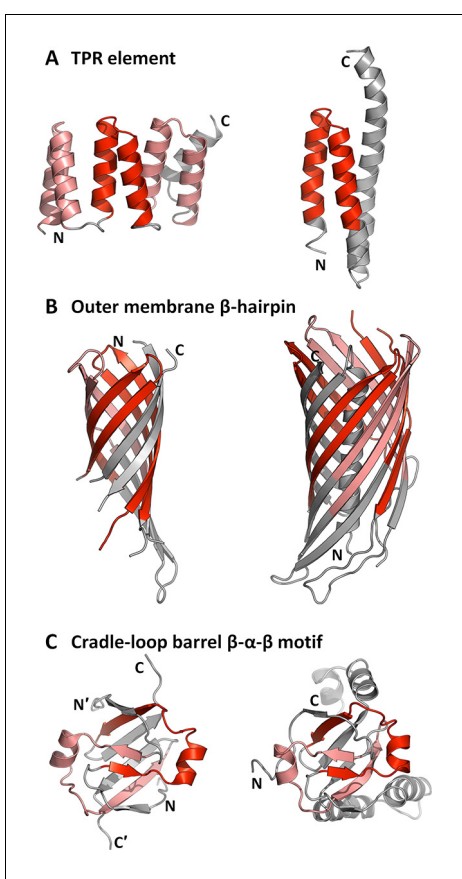

**Figure 7.** Amplification and accretion are key forces in the emergence of domains. Of the 40 fragments in our set, 14 form folds by repetition. The fragments are coloured in red in the shown structures. (**A**) The TPR element (*Figure 3*: 28) occurs repetitively in the TPR-like superfamily (a.118.8; 1ELW, shown on the left side) and singly in six other folds (e.g., a.7.16, 2CRB, right). (**B**) Outer membrane β-barrels comprise 4–12 homologous copies of a β-hairpin element (*Figure 3*: 25); examples include the eight-stranded OmpA (1QJP, left) and the twelve-stranded NalP (1UYN, right). The entire barrels are formed by repetition, but the strands of the hairpin split by the N- and C- termini are left gray. (**C**) The transcription factor AbrB (1YFB, left) is a homodimer and contains one copy of the β-α-β motif (*Figure 3*: 7) per subunit. MraZ has internal sequence symmetry and contains two homologous copies of the β-α-β motif (1N0E, right).

fold, but never in single copy. These are the outer membrane β-hairpin (*Figure 3*: 25) (*Remmert et al., 2010*), which forms β-barrels of between four and twelve hairpins (*Figure 7B*), and the four-stranded β-meander (*Figure 3*: 13) (*Chaudhuri et al., 2008*; *Kopec and Lupas, 2013*), which forms propeller-like toroids containing between four and twelve copies. The remaining two fragments always form folds by duplication (or homo-oligomerization). For example, the cradle-loop barrels, whose evolution we have studied in detail (*Coles et al., 1999*; *2005*; *2006*; *2007*; *2008*), encompass a broad range of topological variants, which are however all built from two copies of a β-α-β fragment (*Figure 3*: 7, *Figure 7C*).

Our findings thus indicate that repetition and accretion must have been the key forces in the emergence of domains. The importance of repetition has been pointed out by many earlier studies. Starting in the 1970's Andrew McLachlan charted out in more than 20 publications the origin of many proteins by repetition (*McLachlan, 1972*; *1987*), in some cases clearly of subdomain-sized fragments (*Blundell et al., 1979*; *McLachlan, 1980*; *1980*). Aided by computational tools to detect repeats in sequence (*Heger and Holm, 2000*; *Szklarczyk and Heringa, 2004*; *Biegert and Soding, 2008*) and structure (*Kim et al., 2010*; *Myers-Turnbull et al., 2014*), analyses of fold space have shown the high incidence of repetitive folds, amounting to as many as one fifth of all folds (*Andrade et al., 2001*; *Balaji, 2015*; *Forrest, 2015*). These include some of the most frequent folds,

such as ferredoxins (*Eck and Dayhoff, 1966*), immunoglobulins (*Huang and Xiao, 2007*), β-propellers (*Murzin, 1992*) and TIM-barrels (*Soding et al., 2006*); indeed, of the 10 most populated folds in SCOPe, 6 (including the top 5), have repetitive structures and all of them have members in which the repetition is also detectable at the sequence level. In most cases, the repetitive structure of these folds has been interpreted as evidence for their origin by amplification of a subdomain-sized fragment and in some cases, such as for β-propellers (*Yadid et al., 2010*; *Voet et al., 2014*), β-trefoils (*Lee and Blaber, 2011*; *Broom et al., 2012*), and TIM-barrels (*Bharat et al., 2008*; *Richter et al., 2010*), this process has also been explored by protein engineering. Only a subset of these fragments that form folds by repetition are present in our set, since we require them to occur in at least two folds to consider them antecedent to folded proteins, but many are currently only detectable in one fold.

### Primordial peptides in proteins today

Our fragments are spread across a total of 130 folds and 188 superfamilies in SCOPe. Given that the current version of SCOPe comprises 1194 folds and 1961 superfamilies, they could be considered to cover only a small of fraction. However, of the 25 most populated folds in SCOPe, which comprise about 25% of all superfamilies, 14 contain one of the fragments and 7 of the top 10 do. We conclude that, far from being anecdotal, our fragments are indeed widespread in today's domains.

The 40 fragments we describe clearly represent a lower bound, given the stringent significance cut-offs in our study, the as yet incomplete fold assignment for known structures, and the fact that some ancestral fragments may have survived only in a single fold and are thus invisible to our method. Indeed, our goal in this study was not completeness, given the extensive overlap in statistical scores that can be expected between the least likely false negatives and the most likely true positives. Rather, it was to assemble a comprehensive set of confident positives in order to show that a signal of homology that predates folding as a widespread property is still detectable in proteins today.

How many fragments may we be missing? With respect to the significance cut-offs we find, for example, that relaxing the sequence similarity requirement to a probability of 60% leads to the inclusion of 5 further fragments, some with substantial arguments in their favor, such as the aforementioned ASP-box (*Copley et al., 2001*) (*Figure 3—figure supplement 1*: B1, *Figure 3—source data 1*). Also, the availability of more sequences and structures, particularly for proteins from poorly sampled branches of life, should allow to bridge more widely separated folds by increasing the significance of their matches. For the fragments that occur in a single fold, we note that because of the importance of repetition, it might be possible to reconstruct further fragments by specifically analyzing repetitive folds. Certainly some of the chemical activities provided by these folds appear sufficiently ancient to justify the expectation that they were covered early in the genesis of folded proteins (*Farias-Rico et al., 2014*; *Lupas et al., 2015*).

For these reasons we think that the number of ancestral fragments that could be reconstructed at a satisfactory level of confidence may well approach 100 in the next decades. Nevertheless, the set of 40 fragments we have described here should cover both the most frequent and best conserved ancestral peptides, and support a viable theory for the emergence of folded proteins.

## Materials and methods

### Detection of sequence- and structure-similar fragments

To assemble domains representative of all known fold types, we chose the SCOPe database (release 2.03) (*Fox et al., 2014*) and filtered it to a maximum of 30% sequence identity, obtaining 9452 domains. Multiple alignments were built for each of these domains using the buildali.pl script (with default parameters) from the HHsearch package (*Soding, 2005*). This script uses PSI-BLAST (*Altschul et al., 1997*) and contains heuristics to reduce the inclusion of nonhomologous sequence segments at the ends of PSI-BLAST sequence matches, the leading cause of high-scoring false positive matches. We used PSI-BLAST, rather than the more sensitive HHblits (*Remmert et al., 2012*), because in our experience the sensitivity of HHblits leads it to occasionally assign elevated probabilities to analogous matches. Profile HMMs were calculated from the alignments using hhmake, also from the HHsearch package, and subjected to pairwise comparisons with HHsearch. Comparisons

were thus always made with the full domains, never with fragments thereof. We used default settings, but switched off secondary structure scoring (option ssm 0) in order to reduce the likelihood that matches were scored highly because of a chance similarity of their (predicted) secondary structures. The HHsearch probabilities we obtained are therefore conservative with respect to the ones obtained for example from the HHpred server in default settings (*Soding et al., 2005*), which is the most frequent source of HHsearch-based deep homology analyses in publications today. We only considered reciprocal matches (of the form domain A matches domain B *and* domain B matches domain A), and assigned these the average of the two obtained HHsearch probabilities. The structures of the aligned segments of SCOPe30 domains were subsequently compared using TM-align (*Zhang and Skolnick, 2005*). We filtered out all matches in which the aligned segment involved only a single secondary structure element.

To establish cut-offs for the comparison of domains of different fold, we plotted comparisons of domains within families (presumed homologs) (*Figure 2A*). The plot shows a bimodal distribution, with the highest representations at the top and bottom. At HHsearch probabilities of $\geq 70\%$ and TM-scores of $\geq 0.5$, about a fourth of all homologous relationships in SCOPe30 are filtered out. We used these cut-offs for the comparison of domains of different fold, as it gives us a substantial margin of safety in the interpretation of relationships presumed analogous by SCOPe.

We gathered all matches between domains of different fold with an HHsearch probability of $\geq 70\%$ and a TM-score of $\geq 0.5$. Next, we eliminated all matches between larger superfamilies that nevertheless only made a single connection. This was done in order to minimize the number of potential false positives. We then applied single-linkage clustering to this resulting set of matches. Fragments from matches between folds were pooled together if they overlapped by at least 80% of their length, i.e. domains A, B, and C were combined together if domain A matched domain B *and* domain B matched domain C such that the boundaries in B overlapped by at least 80%. The resulting clusters were analyzed interactively and merged together if their similarity was based on the presence of a shared sequence- and structure-similar fragment. This yielded a final set of 64 potentially interesting clusters. We generated sequence and structure alignments for the fragments that formed the basis of these clusters and assigned boundaries by manual inspection.

## Smotifs reference set

To obtain a reference set of fragments for our study, we assembled the 40 most frequent supersecondary structure motifs seen in proteins, according to the Smotifs database (*Fernandez-Fuentes et al., 2010*). This is an exhaustive library of geometrically defined local supersecondary structure motifs, composed of two consecutive secondary structures connected by a loop. In this library, motifs of varying lengths, but with similar geometry and secondary structures, are grouped together into clusters, 2296 in total. The Smotifs library is available as a MySQL database; however, the frequency of occurrence of the motif clusters cannot be directly calculated from this database. We were therefore kindly provided with the frequency table by Andras Fiser (Albert Einstein College of Medicine, New York). Using this table, we randomly selected one representative each for the 50 most frequent motif clusters; we required the representative to comprise 24 residues, the median length of our fragments, with at least 6 residues in each secondary structure element and a connecting loop of at most 6 residues. Next, we inspected these representatives and picked the 40 most frequent fragments after eliminating non-compact ones. Finally, for each of these 40 fragments, we picked a prototype in the SCOPe30 database by performing structure searches using TM-align and by selecting the match with the best TM-score. These 40 fragments formed our reference set.

## Correlation of structure and sequence similarity

To evaluate whether the sequence similarity shown by our fragments could be the result of structural convergence, rather than origin from a common ancestor, we searched the SCOPe30 database with each fragment from our set and the Smotifs reference set using TM-align at a cut-off of 0.5. Since our 40 fragments are structurally very similar in all their respective embodiments, we randomly picked one representative of each type. To calculate a sequence-versus-structure plot for our set and the Smotifs set, we followed the methodology described by us in previous studies (*Remmert et al., 2010*; *Kopec and Lupas, 2013*). We used TM-align to compare the Smotifs set with domains from within the respective superfamily (presumed homologous set) and with the

background set that comprised all other folds (presumed analogous set) to search for structurally similar fragments. For our fragments, we compared each representative fragment with domains of all other superfamilies in which we had detected it (homologous) and with all other folds (analogous). In both cases, the TM-score was normalized based on the length of the query motifs. For each structure match, we calculated the profile-profile sequence similarity score with HHalign from the HHsearch package (*Soding, 2005*), however based on the fixed structural alignment obtained from TM-align. The HHalign score was normalized based on the number of aligned residues. Next, each pair of structure and sequence scores was plotted in a scatter plot. To calculate the correlation between TM- and HHalign-scores, we assumed a linear dependency between them and performed linear regression using SciPy (*Jones et al., 2001*). We performed a t-test to determine whether the slope of the regression line differs significantly from zero; we chose a significance level of 1e-10.

## Identification of interactions with nucleic acids, metals, iron-sulfur clusters, nucleotides and nucleotide-derived cofactors

First, we searched the SCOPe30 database for occurrences of each of the 40 reference Smotifs using TM-align at a TM-score cut-off of 0.5 and length coverage of 100%. This yielded occurrences in 323 folds and 447 superfamilies. For comparison, our 40 fragments occur in 130 folds and 188 superfamilies. Following this step, for all occurrences of our fragments and the Smotifs fragments in SCOPe30, we searched for potential interactions with nucleic acids, metals, iron-sulfur clusters, nucleotides, and nucleotide-derived cofactors. A fragment was deemed to interact with these molecules if it made at least three inter-atomic contacts to them within a distance cut-off of 3Å. In addition, we included one of our fragments, the β-α-β motif seen in cradle-loop barrels (*Alva et al., 2008*) (*Figure 3*: 7) into the nucleic acid binding set, even though only a high-resolution model of its interaction with DNA is currently available, based on NMR data (*Zorzini et al., 2015*). Even discounting this addition, the 12 other nucleic acid-binding fragments of our set substantially exceed the two nucleic acid binders found in the Smotifs set, even though these cover about three times as many superfamilies and folds.

## Acknowledgements

This work would not have been possible without Rob Russell (University of Heidelberg) and Chris Ponting (University of Oxford), with whom we framed the original hypothesis for the origin of folded proteins from subdomain-sized peptides. We thank Dek Woolfson (University of Bristol), Nick Grishin (University of Texas Southwestern Medical Center), Kris Brown (GlaxoSmithKline), and members of the Department of Protein Evolution (MPI for Developmental Biology, Tübingen) for many discussions over the years. We are grateful to Andras Fiser (Albert Einstein College of Medicine) for sharing his current Smotifs database prior to publication. Nir Ben-Tal, Philip Bourne, Janusz Bujinicki, Stanislaw Dunin-Horkawicz, Nick Grishin, Rachel Kolodny, Chris Ponting, Rob Russell, Ceslovas Venclovas, Dek Woolfson, and the reviewers of this article are gratefully acknowledged for their comments and suggestions. This work was supported by institutional funds of the Max Planck Society. JS acknowledges support by the German Federal Ministry of Education and Research (BMBF) within the framework of e:Med (grant e:AtheroSysMed, 01ZX1313A-2014) and e:bio (grant SysCore).

## Additional information

### Funding

| Funder | Grant reference number | Author |
|---|---|---|
| Max-Planck-Gesellschaft | | Vikram Alva Johannes Söding Andrei N Lupas |
| German Federal Ministry of Education and Research | e:Med (grant e: AtheroSysMed, 01ZX1313A-2014) and e:bio (grant SysCore) | Johannes Söding |

The funders had no role in study design, data collection and interpretation, or the decision to submit the work for publication.

## Author contributions
VA, Conception and design, Acquisition of data, Analysis and interpretation of data, Drafting or revising the article; JS, Conception and design, Analysis and interpretation of data; ANL, Conception and design, Analysis and interpretation of data, Drafting or revising the article

## Author ORCIDs
Johannes Söding, http://orcid.org/0000-0001-9642-8244
Andrei N Lupas, http://orcid.org/0000-0002-1959-4836

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
