## [Decision Letter]

Thank you for submitting your work entitled "A vocabulary of ancient peptides at the origin of folded proteins" for peer review at *eLife*. Your submission has been favorably evaluated by John Kuriyan (Senior editor) and three reviewers.

The following individual responsible for the peer review of your submission has agreed to reveal his identity: Rob Russell (peer reviewer).

The reviewers have discussed the reviews with one another and the Reviewing editor has drafted this decision to help you prepare a revised submission.

Review:

In this most interesting study, Lupas and colleagues hypothesize that modern protein domains arose from ancestral peptides. This is not a new idea; however, the current work is a systematic study of many proteins, while previous work focused on individual fold families. The authors have systematically revisited the original "antecedent domain segment" notion for protein structure, published some 15 years ago, by now using a systematic interrogation of structure and sequence. The authors identify 40 short 3D-structural motifs (they call them "fragments") that are most likely to be some of the ancestral primordial peptides that gave rise to the present-day protein world. These are subdomain-size segments found by sequence profile similarity and validated by 3D structure similarity. They typically lack a well-defined hydrophobic core and are complemented to form complete domains in several different and likely evolutionarily independent ways.

This work stems from a hypothesis that proteins originated by fusion of ancestral peptides. These peptides likely functioned in complex with RNA. The crux of the authors' argument is that subdomain sequence similarity between different domains indicates deep homology for the subdomain. Because the containing domains are not homologous, they argue that the homology at the subdomain level indicates an ancestral peptide shared by the domains. They further argue that this sequence similarity does not arise from structural constraints on sequence, as similar subdomain structures with apparently unrelated sequences can be found. The authors' arguments are mostly clear. They also do not overstate their conclusion that these peptides are ancestral, which necessarily remains somewhat speculative.

This study is made possible by the development of the HHsearch algorithm to find statistically supported similarity between sequence profiles. While some of the HHsearch hits represent false positives, most are predictive of 3D structure, as has been validated in many CASP experiments. It is indeed most likely that if the HHsearch sequence-based alignment is nearly the same as the 3D structure-based alignment then the two segments are homologous. This is the main assumption of this work.

Some of these fragments have been detected before and studied, e.g. in the SISYPHUS database and various publications (e.g. about HHH motifs), but this manuscript provides the first principled and comprehensive approach to find them all and therefore deserves attention. Interestingly, no example of multiple different peptides in any fold was found, which goes slightly against the original idea, though multiplication/repetition of fragments seems definitely to have been a theme in the evolution of the folds.

In summary, the elaboration presented in this paper of the idea that short sub-structure motifs might have arisen through primordial interactions with RNA will be of general interest to the readership of *eLife*, and the paper is potentially suitable for publication provided that the authors can address the following issues raised by the referees.

Important issues to address:

1) There is concern about the argument that the sequence correlation between these motifs must arise from history rather than convergence. This argument – the core argument in the paper – hinges on the idea that the sequence space compatible with the subdomain motif is too vast to lead to chance convergence. The authors do not, however, provide any estimate for the number of sequences compatible with a given subdomain motif. Their median fragment length is 24 residues. How many different amino acids can be tolerated at each site in the fragment? With what probability? If only a few amino acids can be tolerated at each position, the space associated with a 24 residue fragment could be quite small, potentially leading to a spurious signal of homology.

The reviewers wonder whether it would be feasible to compute more of a significance of the overall observation (i.e. considering whether such a distribution of sequence similarities might be expected by chance) than just what seems to be a rather lenient sequence similarity measure. The reviewers are concerned that the authors support the lack of convergence with anecdotal rather than statistical analysis.

As an aside, the reviewers recognize that a compelling point is that these ancient peptide candidates are associated with ancient functions, and perhaps this isn't emphasized enough in the paper.

2) One thing that is missing is more discussion of the fragments that were common, but not significant in terms of sequence. For instance, the point about there being no single fold containing two fragments leads to the question of just how many there were that satisfied only stringent structural criteria. The reviewers realise that this is considerable work, but wonder if the authors have these data lying around anyway. For example, of the most popular folds, how many have two common fragments or more, even if HMMsearch doesn't give a significance? This would lend some insight into the really ancient, no-longer-detectable, but still related relationships.

3) Some discussion of some old favorites is also missing. The two halves of the Immunoglobulin fold or Rossmann fold, etc. – that is the cases that most structurally-obsessed readers will have in their minds, could also, if fitting, do with some discussion in the same light.

Other issues to address:

1) The authors can reference more relevant publications (e.g. SISYPHUS and those dedicated to individual motifs, like HHH).

2) The description of your methods to establish the statistical cutoffs for sequence and structural similarity took multiple reads to understand (both in the text, Results and discussion, and in the graphic and legend of Figure 2). Highlighting the fragments on Figure 2 might help.

3) "domains might not constitute the evolutionary unit of protein structure" might better be stated as "domains might not constitute the only evolutionary unit of protein structure"

---

## [Author Response]

*Important issues to address:*

*1) There is concern about the argument that the sequence correlation between these motifs must arise from history rather than convergence. This argument – the core argument in the paper – hinges on the idea that the sequence space compatible with the subdomain motif is too vast to lead to chance convergence. The authors do not, however, provide any estimate for the number of sequences compatible with a given subdomain motif. Their median fragment length is 24 residues. How many different amino acids can be tolerated at each site in the fragment? With what probability? If only a few amino acids can be tolerated at each position, the space associated with a 24 residue fragment could be quite small, potentially leading to a spurious signal of homology. The reviewers wonder whether it would be feasible to compute more of a significance of the overall observation (i.e. considering whether such a distribution of sequence similarities might be expected by chance) than just what seems to be a rather lenient sequence similarity measure. The reviewers are concerned that the authors support the lack of convergence with anecdotal rather than statistical analysis. As an aside, the reviewers recognize that a compelling point is that these ancient peptide candidates are associated with ancient functions, and perhaps this isn't emphasized enough in the paper.*

We agree that the issue of homology vs. analogy of our fragments is of overriding importance to the manuscript and we would first like to address the question of whether our sequence similarity measures are too lenient to provide sufficient confidence in our inference of homology.

As we noted in the last paragraph of the section on “Reconstructing a vocabulary of primordial peptides”, molecular homology is an unprovable proposition. It can only be approached by probabilistic arguments, which are constantly renegotiated in the scientific community. In our experience, this renegotiation has consistently admitted as evidence increasingly distant sequence matches made with new tools, after a period of doubt. Some examples that we have witnessed, where today the sequence evidence is considered conclusive (and we feel sure that the reviewers could report similar examples): The 1985 paper reporting the homology of response regulators was rejected by *Science* because FASTP scores could not be taken as evidence of homology; the paper was eventually published using the member track at PNAS. A 1998 paper delineating the superfamily of 8-stranded OMPs encountered similar criticism aimed at PSI-Blast, but was eventually published, as was a 2010 paper describing the homology of SMP, BPI and Takeout proteins, where HHpred connections at 50-80% were considered inconclusive, even when made multiply across the superfamilies. After an initial lag in acceptance, all these methods became standard applications for homology searches at significance cutoffs that just a few years earlier had seemed too lenient to be significant. Today, HHpred has become the method of choice for detecting deep homology and is used in hundreds of publications each year at more lenient cutoffs than those that we had such problems getting past the reviewers in 2010.

The question is thus whether our approach is still within the currently accepted boundaries for inferring homology, or pushes into new territory. As evidence that we are in fact within the currently accepted boundaries, we place great importance on the observation that our approach systematically recapitulates all reports of common evolutionary fragments made individually over more than 40 years. This covers 40% of our fragments. While we agree that it does not represent statistical analysis, we think that it goes well beyond anecdotal support. We can further substantiate that we stay well within these boundaries by comparing our use of HHsearch to what is already common usage in a wide range of publications. This comparison shows that our use is more conservative than the current standard. Thus:

We made all searches with full domains, never with fragments thereof;

We turned off secondary structure scoring;

We built the profile HMMs with PSI-Blast, not with the more sensitive HHblits.

Given these constraints, the HHsearch probabilities we use are much more stringent than the same probabilities obtained in default settings from the HHpred server (the most common source of HH searches). Indeed, their stringent nature can be seen by the large number of intrafamily matches (i.e. “safe” homologs) falling beneath the threshold in our analysis of the SCOP database (Figure 2).

We further raised the bar for observed matches by requiring that:

They be above our thresholds in both directions of the comparison between two superfamilies of different fold;

They cover more than a single secondary structure;

They be supported by at least two reciprocal connections made between different families within the superfamily (provided the superfamily contained two families or more).

Given all these restrictions, we would argue that our procedure is stringent, rather than lenient, with respect to the procedures commonly used today in publications that describe deep homology.

We did consider carefully avenues for statistical support of our argument prior to submission and all the more so after receiving the reviewers’ comments. Based on our attempts, we do not think, however, that the approach proposed by the reviewers would work in a satisfactory fashion.

For an estimate of the number of sequences compatible with a given fragment we would need to solve the reverse protein folding problem – not a feasible proposition. Failing that, we could try an approximation, identifying a given fragment by structure comparisons in known protein structures and tabulating a frequency matrix of amino acids at the individual positions. The quality of this approximation would depend on the number of structurally similar fragments identifiable outside our presumed homologous set. We have explored this for a few of our fragments and estimate that SCOPe30 would yield less than 10^[2]^ instances even for fragments corresponding to common supersecondary structures, such as α- or β-hairpins. For fragments that are uncommon, there would be correspondingly fewer instances outside our homologous set, in some cases (such as for fragments 6, 9, 10, and 11) there would be none. If we scaled our search up to the larger SCOPe95 (a laborious endeavor), we would still not exceed numbers in the low hundreds for all but a few fragments; scaling up further to PDB would be unfeasible, as we would have to separate the structures into domains and classify them in order to be able to judge whether they should be in the homologous or analogous set for any given fragment.

Thus, even if we were able to implement this approach, the low number of instances would doom the approximation. At the restricted number of occurrences available, it would be impossible to decide whether a given fragment *could* not have been made with a broader sequence representation, or simply had not been explored by nature in more than a few folds. It is well established that nature has not explored more than an infinitesimal part of sequence space at the domain level; the extent to which it has done so at the supersecondary structure-level has not been established so far.

For these reasons, our statement that almost all of our fragments have matches with substantial structural similarity, but undetectable sequence similarity, in SCOPe30 is indeed anecdotal evidence that these fragments can be made in more than one way. This is however just the preamble to a statistical analysis, described in the same paragraph and Figure 5. This analysis, tracking the correlation between sequence and structure similarity in presumed homologs vs. presumed analogs, is much less sensitive to a small number of instances for any given fragment and can be applied collectively to all fragments. A correlation between sequence and structure similarity should not be detectable in analogs, if structural constraints do not result in a sequence bias. It should however be detectable in homologs, as these would have ancestrally started with identical sequences and structures, decaying in time towards the baseline.

As a control database for this analysis we used 40 of the most frequent supersecondary structures in PDB (from the Smotifs dataset), identified them in SCOPe30 and plotted their pairwise sequence and structure similarities, separately for matches within superfamilies (presumed homologs; red) and matches between folds (presumed analogs; blue). The result confirms our expectation that for these fragments, sequence bias resulting from structural constraints is very low, as judged from the very low correlation in the analogous set (r = 0.12); certainly it disproves the notion that only a few amino acids could be tolerated at each position of a fragment. In contrast, again as expected, homologs show a clear correlation (r = 0.56).

Performing the same analysis for our 40 fragments yields substantially the same result. Matches outside our list of postulated homologs (blue) show as little correlation between sequence and structure as the Smotifs set (r = 0.14), emphasizing the very low sequence bias introduced by structural constraints. Matches between our proposed homologs, however, show a similar correlation as the SCOP homologs (r = 0.38), even though our comparisons are made across a wider evolutionary distance (between superfamilies) than the reference comparisons (within superfamilies).

In conclusion, we think that our analysis of sequence-structure correlation can discriminate between homology and analogy, and directly addresses the question raised by the referees as to whether structural constraints may have caused the sequence similarity of our fragments: They have not.

We have attempted to expand the points made here in the manuscript and make our reasoning more explicit in the text and the Materials and methods; we hope that our arguments have become clearer as a result.

*2) One thing that is missing is more discussion of the fragments that were common, but not significant in terms of sequence. For instance, the point about there being no single fold containing two fragments leads to the question of just how many there were that satisfied only stringent structural criteria. The reviewers realise that this is considerable work, but wonder if the authors have these data lying around anyway. For example, of the most popular folds, how many have two common fragments or more, even if HMMsearch doesn't give a significance? This would lend some insight into the really ancient, no-longer-detectable, but still related relationships.*

This is an excellent point, given our conjecture that fragment combinations “might lead to the retention of only one dominant fragment, the other(s) adjusting by rapid divergence, making them undetectable by our methods”. We were happy to follow it up. Since these data were not available and the effort is indeed not trivial, we only selected one family for each superfamily that contains one of our fragments (188 in all) for the purposes of this revision, but we intend to do this analysis comprehensively in the future.

For combinations between a “significant” and a “nonsignificant” fragment, we first asked that the second fragment have a TM-score ≥ 0.5 and an HHsearch probability ≥ 50%, in order to see whether there were any fragment combinations that had remained just under our sequence cutoff. There were none. We then relaxed the sequence cutoff to HHsearch probabilities ≥ 10%, but again there were no combinations apparent. Only when we omitted sequence cutoffs entirely for the second fragment did we find combinations and these were in fact quite frequent, arguing that the lack of compatibility was not due to geometric considerations. We now explicitly address this point in the paper and have generated a new figure to show some of the most attractive instances of fragment combinations.

*3) Some discussion of some old favorites is also missing. The two halves of the Immunoglobulin fold or Rossmann fold, etc. – that is the cases that most structurally-obsessed readers will have in their minds, could also, if fitting, do with some discussion in the same light.*

Agreed. We had provided only a cursory reference to the broad range of studies exploring internal symmetry in domains and its potential evolutionary implications, where these had not led to identification of an ancestral fragment by our criteria (last paragraph of the section on “Repetition as a dominant force in the origin of folds”). For several of these, however, ancestrality could well be argued, as we state in the last paragraph of the manuscript (“For the fragments that occur in a single fold, we note that because of the importance of repetition, it might be possible to reconstruct further fragments by specifically analyzing repetitive folds.”).

We have now expanded this point in the manuscript at the abovenamed locations.

*Other issues to address:*

*1) The authors can do a better job referencing relevant publications (e.g. SISYPHUS and those dedicated to individual motifs, like HHH).*

Agreed and done. We have added more than 40 references in revision and hope to have thus provided a broader context to our manuscript.

*2) The description of your methods to establish the statistical cutoffs for sequence and structural similarity took multiple reads to understand (both in the text, Results and discussion, and in the graphic and legend of Figure 2). Highlighting the fragments on Figure 2 might help.*

We have hopefully been able to clarify this point in the text and the figure legend. Unfortunately, mapping our fragments onto Figure 2 would be a major undertaking, so we have not attempted it for this revision.

*3) "domains might not constitute the evolutionary unit of protein structure" might better be stated as "domains might not constitute the only evolutionary unit of protein structure"*

Done.